# Genetic diversity and neutral selection in *Plasmodium vivax* erythrocyte binding protein correlates with patient antigenicity

Jin-Hee Han[1,2]*, Jee-Sun Cho[3,4], Jessica J. Y. Ong[1], Ji-Hoon Park[2], Myat Htut Nyunt[5], Edwin Sutanto[6], Hidayat Trimarsanto[6], Beyene Petros[7], Abraham Aseffa[8], Sisay Getachew[7,8], Kanlaya Sriprawat[9], Nicholas M. Anstey[10], Matthew J. Grigg[10,11], Bridget E. Barber[10,11], Timothy William[11,12,13], Gao Qi[14], Yaobao Liu[14,15], Richard D. Pearson[16,17], Sarah Auburn[10,18,19], Ric N. Price[10,18,19], Francois Nosten[9,18], Laurent Rénia[3], Bruce Russell[1], Eun-Taek Han[2]*

1 Department of Microbiology and Immunology, University of Otago, Dunedin, New Zealand, 2 Department of Medical Environmental Biology and Tropical Medicine, School of Medicine, Kangwon National University, Chuncheon, Gangwon-do, Republic of Korea, 3 Singapore Immunology Network (SIgN), Agency for Science, Technology and Research (A*STAR), Singapore, 4 Jenner Institute, Old Road Campus Research Building, Roosevelt Drive, Oxford, United Kingdom, 5 Department of Medical Research, Yangon, Myanmar, 6 Eijkman Institute for Molecular Biology, Jakarta, Indonesia, 7 College of Natural Sciences, Addis Ababa University, Addis Ababa, Ethiopia, 8 Armauer Hansen Research Institute, Jimma Road, Addis Ababa, Ethiopia, 9 Shoklo Malaria Research Unit, Mahidol-Oxford Tropical Medicine Research Unit, Faculty of Tropical Medicine, Mahidol University, Mae Sot, Thailand, 10 Global and Tropical Health Division, Menzies School of Health Research and Charles Darwin University, Darwin, Australia, 11 Infectious Diseases Society Sabah-Menzies School of Health Research Clinical Research Unit, Sabah, Malaysia, 12 Clinical Research Centre, Queen Elizabeth Hospital, Sabah, Malaysia, 13 Gleneagles Hospital, Sabah, Malaysia, 14 Jiangsu Institute of Parasitic Diseases, Wuxi, Jiangsu, People's Republic of China, 15 Medical College of Soochow University, Suzhou, Jiangsu, People's Republic of China, 16 Big Data Institute, Li Ka Shing Centre for Health Information and Discovery, Old Road Campus, Oxford, United Kingdom, 17 Wellcome Sanger Institute, Hinxton, Cambridge, United Kingdom, 18 Centre for Tropical Medicine and Global Health, Nuffield Department of Clinical Medicine Research Building, University of Oxford Old Road Campus, Oxford, United Kingdom, 19 Mahidol-Oxford Tropical Medicine Research Unit, Faculty of Tropical Medicine, Mahidol University, Bangkok, Thailand

* han.han@kangwon.ac.kr (JHH); etaekhan@gmail.com (ETH)

**Data Availability Statement:** All relevant data are within the manuscript and its Supporting Information files.

## Abstract

*Plasmodium vivax* is the most widespread and difficult to treat cause of human malaria. The development of vaccines against the blood stages of *P. vivax* remains a key objective for the control and elimination of vivax malaria. Erythrocyte binding-like (EBL) protein family members such as Duffy binding protein (PvDBP) are of critical importance to erythrocyte invasion and have been the major target for vivax malaria vaccine development. In this study, we focus on another member of EBL protein family, *P. vivax* erythrocyte binding protein (PvEBP). PvEBP was first identified in Cambodian (C127) field isolates and has subsequently been showed its preferences for binding reticulocytes which is directly inhibited by antibodies. We analysed PvEBP sequence from 316 vivax clinical isolates from eight countries including China ($n = 4$), Ethiopia ($n = 24$), Malaysia ($n = 53$), Myanmar ($n = 10$), Papua New Guinea ($n = 16$), Republic of Korea ($n = 10$), Thailand ($n = 174$), and Vietnam ($n = 25$). PvEBP gene exhibited four different phenotypic clusters based on the insertion/deletion (indels) variation. PvEBP-RII (179–479 aa.) showed highest polymorphism similar to other

**Funding:** This study was supported by the B.R. laboratory from the Marsden Fund 17-UOO-241 (BR) and by the National Research Foundation of Korea (NRF) grant funded by the Korea government (MSIP) (NRF-2017R1A2A2A05069562) (E-T.H), by the Basic Science Research Program through the National Research Foundation of Korea (NRF) funded by the Ministry of Science, ICT and Future Planning (2015R1A4A1038666) (E-T.H), N.M.A and M.J.G supported by National Health and Medical Research Council Fellowships. J.S.C held a Singapore International Graduate Award (SINGA). R.N.P. is a Wellcome Trust Senior Fellow in Clinical Science (200909). The funders had no role in study design, data collection and analysis, decision to publish, or preparation of the manuscript.

EBL family proteins in various *Plasmodium* species. Whereas even though PvEBP-RIII-V (480–690 aa.) was the most conserved domain, that showed strong neutral selection pressure for gene purifying with significant population expansion. Antigenicity of both of PvEBP-RII (16.1%) and PvEBP-RIII-V (21.5%) domains were comparatively lower than other *P. vivax* antigen which expected antigens associated with merozoite invasion. Total IgG recognition level of PvEBP-RII was stronger than PvEBP-RIII-V domain, whereas total IgG inducing level was stronger in PvEBP-RIII-V domain. These results suggest that PvEBP-RII is mainly recognized by natural IgG for innate protection, whereas PvEBP-RIII-V stimulates IgG production activity by B-cell for acquired immunity. Overall, the low antigenicity of both regions in patients with vivax malaria likely reflects genetic polymorphism for strong positive selection in PvEBP-RII and purifying selection in PvEBP-RIII-V domain. These observations pose challenging questions to the selection of EBP and point out the importance of immune pressure and polymorphism required for inclusion of PvEBP as a vaccine candidate.

## Author summary

When developing a malaria vaccine, it is essential to consider natural polymorphisms of the candidate antigen to ensure high efficacy. As a novel member of EBL protein family in *P. vivax*, PvEBP showed preference for reticulocyte binding, with its specific antibody exhibiting binding inhibition activity. This study presents PvEBP as a suitable target for an asexual erythrocytic stage vaccine. Here, we discuss genetic polymorphisms and neutral selection of PvEBP gene in eight different *P. vivax*-endemic countries, and how these affect the prevalence of naturally-acquired anti-PvEBP antibodies from vivax patients. This study highlights a number of challenges associated with the PvEBP base vaccine development strategy.

## Introduction

In 2017 *Plasmodium vivax* caused between 7.5–14.3 million cases of malaria, mainly in the South-East Asia region (56%) [1, 2]. Although *P. vivax* is generally not lethal to their host, *P. vivax* causes high morbidity among the five human invasive *Plasmodium* species (*P. falciparum*, *P. vivax*, *P. knowlesi*, *P. malariae* and *P. ovale*) due to recurrent parasitaemia from reactivation of the dormant hypnozoites [1, 3–5]. Despite its importance as the most widespread *Plasmodium* species, absence of reliable *in vitro* long-term culture system has impeded research into optimal interventions against the parasite [5, 6]. Current malaria vaccine development strategies focus on identifying a specific, immunogenic antigen which will stimulate protective humoral immune response and to produce sufficient amount of the specific, functional antibody to provide sterile immunity against malaria infection. Finding such functional antibodies from individuals living in endemic settings may answer for natural infections against malaria. Anti-malarial humoral immune responses provide various function including phagocytosis and/or direct killing by complement mediation [7] which results in a reduction in merozoite invasion, growth and rosetting formation [7–9].

Erythrocyte invasion of *Plasmodium* species occurs by sequential multiple molecule interactions, with each step mediated by antigens belonging to different protein families [10]. The

erythrocyte binding-like (EBL) protein family which contained EBL (or RII) domain was identified in various *Plasmodium* species with conserved function as host cell binding via glycoprotein receptor for invasion before tight junction formation [11, 12]. These proteins are highly expressed in mature schizont stage and localized in the microneme [12]. The best-known member of EBL family is *P. vivax* Duffy binding protein (PvDBP) which binds to Duffy antigen receptor for chemokine (DARC/Fy glycoprotein) and mediate invasion into erythrocytes by the majority of *P. vivax* isolates [13]. This specific ligand-receptor interaction and parasite invasion process can be a target for invasion blocking antibodies against EBL domain (PvDBP-RII) [14, 15]. Given its unique importance, the EBL domain has been the main candidate for vaccine development. However, recent Phase 1a PvDBP-RII vaccine clinical trial showed low efficacy [16] and strain-specific immune response is limited by a high level of genetic polymorphism in the EBL domain [17]. To overcome this important issue, identifying conserved epitopes and evaluating their recognition by neutralising antibodies is essential to achieve sterile immunity [18, 19]. Further investigations of novel antigens for vaccine candidates are required to understand downstream immune response of target antigen, evaluation of current immunity to conserved epitopes, and determination of regions undergoing neutral selection.

Here, we have focused on another member of EBL family, *i.e. P. vivax* erythrocyte binding protein (PvEBP), which was first identified from a Cambodian field isolate (C127) [20]. PvEBP preferentially binds CD71$^+$ and CD234$^+$ reticulocytes but not CD234$^-$ reticulocyte through its EBL domain, PvEBP-RII [21]. Antibodies against PvEBP-RII but not PvDBP is inhibited binding of reticulocytes to PvEBP expressing COS cells indicating that despite sequence similarity the binding regions of these two EBL proteins are different [21]. An increasing number of studies have reported *P. vivax* invasion in Duffy-negative populations in Africa regions highlighting potential of PvDBP independent invasion pathways [22–24]. Given its similarity, it is hypothesised that, PvEBP may have a role in facilitating this newly discovered DARC-independent invasion pathway [21]. In addition to the analysis of PvEBP for its functional contribution to parasitic invasion, there is also a need to evaluate immune responses and natural polymorphisms of EBP in field isolates to further determine the likelihood of EBP as a possible vaccine candidate.

We analysed clinical isolates from eight countries to quantify genetic diversity of PvEBP and total serum concentrations of IgG against two domains of PvEBP (RII and RIII-V). Our analysis evaluates the correlation of vivax patient antigenicity with genetic polymorphism and natural selection to explore the prevalence of PvEBP specific IgG and how this might impact vaccine development.

## Methods

### Ethics statement

Whole blood was collected from symptomatic *P. vivax* patients after examining blood smears using light microscopy at local health centre in five countries for genomic DNA extraction following *pvebp* amplification: Papua New Guinea (PNG, *n* = 16), South region of demilitarized zone in Republic of Korea (ROK, *n* = 10), Shwegyin in Myanmar (*n* = 10), North-western Thailand (*n* = 72), and Vietnam (*n* = 25). Patients sera were collected from three countries: ROK (*n* = 50), Myanmar (*n* = 50), and Thailand (*n* = 37). All clinical samples were collected under the following ethical guidelines and approved protocols: Kangwon National University Hospital Ethical Committee, Republic of Korea (IRB No. 2014-08-008-002), Department of Medical Research, Republic of the Union of Myanmar (Approval No-52/Ethics, 2012), and OXTREC 027–025 (University of Oxford, Centre for Clinical Vaccinology and Tropical

Medicine, UK) and MUTM 2008–215 from the Ethics Committee of Faculty of Tropical Medicine, Mahidol University, Thailand. All adult subjects provided informed written consent, and a parent or guardian of any child participant provided informed written consent on their behalf.

## Genomic DNA extraction, sequencing, and sequence data collection

Genomic DNA was extracted from 200 μl whole blood samples using QIAamp DNA Blood Mini Kit (QIAGEN, Hilden, Germany) following the manufacturer's protocol. Total 133 samples from PNG, ROK, Myanmar, Thailand, and Vietnam were amplified. The primer sets were designed using C127 isolate *pvebp* sequence as a reference (NCBI accession number, KC987954): PvEBP forward (5'-GACTTCCTGACTGGCGTGATTTAC-3') position in 5' UTR region and PvEBP reverse (5'-AGGTATTATCCTCCTAAACAGTTTGTTC-3') position in second intron. The amplicons contained extracellular domain (ecto) and were sequenced using four internal sequencing primer set: fragment 1–2 reverse (5'-AATTTCCATGCGCCACGATGT-3'), fragment 3 forward (5'-ATTCAATAAATGGAAGAAGCATAATAGC-3'), fragment 4 forward (5'-GATCATACTAAAGAAGGAGCAATGG-3'), and fragment 5 forward (5'-CCTACTAATGAGGGTGATAGCGTC-3') using an ABI 3700 Genetic Analyzer (Genotech, Daejeon, ROK). The detailed gene sequences of *pvebp-ecto* are available at GenBank (accession numbers: MN853168—MN853300).

*Pvebp* sequence from China (*n* = 4), Ethiopia (*n* = 24), Malaysia (Sabah, *n* = 53), and Thailand (North-western, *n* = 102) obtained from whole genome sequencing data which described at previously [25–27]. Total 316 sequences were aligned with PNG (PVP01_0102300), Mauritania I (NCBI, bioproject accession: PRJNA67237), and India VII (PRJNA65119) *pvebp* genes as a reference sequence.

## Nucleotide diversity and neutral selection

*Pvebp* nucleotide diversity ($\pi$) is defined as the average number of nucleotide differences per site between two sequences within the sequences. The number of polymorphic sites, number of haplotypes and haplotype diversity (Hd) were calculated by DnaSP software [28]. Test of neutral selection was evaluated using multiple calculation method including Tajima's D, Fu and Li's D*, and Fu and Li's F* with excluding the sequence gap [29, 30]. Under neutrality, Tajima's D is expected to be 0. Significant positive values of Tajima's D indicate population bottlenecks and balancing selection, whereas negative values suggest population expansion or negative selection [29]. Significant positive value of Fu and Li's D* and Fu and Li's F* represent population contraction due to selection. On the other hand, negative values represent population expansion and excess of singletons [30]. Natural selection was determined by calculating the rates of synonymous substitutions per synonymous site ($d_S$) and non-synonymous substitutions per non-synonymous site ($d_N$) at the intra-species level. The calculation was performed by computed Nei and Gojobori's method and robustness was estimated by the bootstrap method with 1000 pseudo replicates as implemented in the MEGA 5 software. A $d_N/d_S$ ratio less than 1 indicates a purifying selection and $d_N/d_S$ ratio greater than 1, indicates a positive selection. The test of *pvebp* natural selection at the inter-species level was performed using the robust McDonald and Kreitman (MK) test with *P. cynomolgi* DBP2 (PcyM_0102400) gene as an out-group using DnaSP software. PvEBP haplotype was evaluated by DnaSP software and graphical presentation for distance in relationship was generated by median-joining method in Network 5.0 software.

## Recombinant protein expression

Recombinant PvEBP-RII (188–421 aa.) and PvEBP-RIII-V (480–676 aa.) fragments were selected using PVP01_0102300 sequence for antigenicity prevalence screening using protein

array. Briefly, PvEBP-RII was cloned into wheat germ cell-free expression vector pEU-E01-His-TEV-MCS (Cell-Free Sciences, Matsuyama, Japan) using In-Fusion HD Cloning kit (Clontech, Mountain View, CA, USA). The domain was amplified using RII specific primer (forward: 5'-gggcggatatctcgagTGTAACGCCAAGAGGGAACGTG-3' and reverse: 5'-gcggta cccgggatccctaGGTATCCCATTGCTCCTTCTTTAG-3'). Lower case letter indicates vector site and underline indicates restriction enzyme site for *XhoI* and *BamHI*. Recombinant PvEB-P-RII was expressed using wheat germ cell-free system (Cell-Free Sciences) following the manufacturer's protocol. RIII-V was cloned into pET28a+ vector for *Escherichia coli* expression using In-Fusion system. RIII-V specific primer (forward: 5'-aatgggtcgcggatccCACAAAGGT GTAAAAATTGCGG-3' and reverse: 5'-ggtggtggtgctcgagTTGCGCATTACTATACCCGTC G-3') amplified region containing *BamHI* and *XhoI* enzyme sites. The plasmid DNA was transformed to BL21 (DE3) (Millipore, Billerica, MA, USA) and expression induced by 0.1 M Isopropyl β-D-1-thiogalactopyranoside (IPTG). The crude proteins were purified by Ni-NTA column (QIAGEN) with 500 mM and 100 mM imidazole, respectively. Each recombinant protein (1 μg/lane) was prepared in 2x reducing buffer and separated by 12% SDS-PAGE. The recombinant protein was visualized with Coomassie brilliant blue staining.

## Antigenicity evaluation

Protein microarray was performed to evaluate total IgG reactivity. 3 aminopropyl-coated slides were prepared as described previously [31]. The slides were printed to each spot with recombinant protein (RII, 200 ng/μl and RIII-V, 12.5 ng/μl) as a saturated concentration and incubated for 2 hours at 37˚C. The recombinant protein coated slide was blocked with blocking buffer (5% BSA in PBS-T) for 1 hour at 37˚C. Vivax patient and healthy individual sera were diluted in PBS-T to 1:25 and probed on the chip for 1 hour at 37˚C. The arrays were visualized with 10 ng/μl of Alexa Fluor 546 goat anti-human IgG (Invitrogen, Carlsbad, CA, USA) in PBS-T for 1 h at 37˚C and scanned with ScanArray Gx laser confocal scanner (PerkinElmer, Norwalk, CT, USA). The positive cut-off values calculated by negative control mean fluorescence intensity (MFI) plus two standard deviations.

## Statistical analysis

The data were analyzed using GraphPad Prism (GraphPad Software, San Diego, CA, USA), SigmaPlot (Systat Software Inc., San Jose, CA, USA), and Microsoft Excel 2013 (Microsoft, Redmond, WA, USA). For the protein array, Student's *t*-test was used to compare the experimentally measured values of each group. The correlation of clinical information with antigenicity was calculated by Pearson correlation test. Differences of $p < 0.05$ were considered significant. The total IgG reactivity index was calculated by each MFI divided by average negative MFI of RII and RIII-V, respectively, for normalization and reactivity comparison between RII and RIII-V.

## Results

### Sequence alignment of *pvebp*

The complete *pvebp* (PVP01_0102300) sequence encodes 2,942 bp with three introns on chromosome 1 in the hypervariable sub-telomeric region [32]. The four exon sequence encodes 2,538 bp nucleotide with 95.1 kDa predicted molecular weights. PvEBP regions are divided based on PvDBP region division strategy in primary structure and they are defined as region I (RI, 1–534 bp), RII (535–1,437 bp), RIII-V (1,438–2,070 bp), and RVI (2,071–2,307 bp) in the first large exon [33]. The RII (EBL domain) and RVI (EBA-175 homologue, or C-terminal

cysteine rich domain) were highly conserved in multiple cysteine residue with *pvdbp*. Additionally, PvEBP contained putative signal peptide (1–60 bp) in RI domain and transmembrane domain (2,326–2,394 bp) in RVII (Fig 1A, PvEBP Cluster 1).

Four ROK isolates (K01, K02, K04, and K05) showed different lengths of *pvebp*-ecto amplicon size (S1 Fig). The sequencing result confirmed 6 nucleotides (GGCAAA) deletion coding 2 amino acids (Gly-Lys/GK) in RII (position in 1,003–1,008 bp), followed by a large insertion in RIII-V (1,564–1,800 bp) comprising of 79 amino acids (Fig 1A and 1B, Cluster 4). From this result, the expanded blast analysis was performed using large insertion sequence for clustering by indels (insertion/deletion) variation type of *pvebp*. Two vivax reference sequences found identical large insertion sequence in Mauritania I (Fig 1A and 1B, Cluster 3) and India VII (Fig 1A and 1B, Cluster 4). Finally, four different *pvebp* cluster identified based on gene phenotypic variation. Almost isolates were classified cluster 1 (304/316, 96.2%) which contained GK without large insertion such as PVP01_0102300 pattern (Fig 1A and 1B, Cluster 1). The cluster 2 (1/316, 0.3%) has GK deletion in RII domain and cluster 3 (7/316, 2.2%) gene phenotype was same with Mauritania I sequence which found in only Ethiopia isolates (Fig 1A and 1B, Cluster 2 and 3). The cluster 4 (4/316, 1.3%) followed India VII sequence as a reference, which found in only ROK isolates (Fig 1A and 1B, Cluster 4). The large identical insertion sequence contained rich Gly (20.3%), Ser (17.7%), and Arg (12.7%) within 79 aa. lengths (Fig 1B). Further analysis using *P. knowlesi* individual Sequence Read Archive (SRA) database on the blast of Pk1 A+, PkNA1, Pk Nuri, PkA1H1 and Pk H (AW) did not yield similar sequences of *ebp* gene. The blast searching with *P. cynomolgi* gave 91% hit identity to *Mulligan* strain Duffy Binding Protein 2 (PcyM DBP2) (Fig 1A, PcyDBP2 M strain). Sequence alignment to PcyM DBP2 identified similar gene phenotypic pattern with PvEBP cluster 4 (Fig 1B). However, PcyM DBP2 presented additional insertion sequence for four amino acid (DERS) upstream of the large insertion, followed by a 20 amino acids expanded insertions (DRSSDGSSGGSSGGSSGGSS) in downstream of large insertion (Fig 1B).

## Genetic diversity and neutral selection of *pvebp*-ecto

*Pvebp*-ecto nucleotide diversity ($\pi$) of 316 isolates from eight countries shown 53 polymorphic sites which included 8 synonymous and 45 non-synonymous sites. Nucleotide diversity based on geographical, China ($\pi \pm$ S.D., 0.00179 ± 0.00046) shown highest nucleotide diversity and followed by Ethiopia (0.00172 ± 0.00011), ROK (0.00169 ± 0.00016), Myanmar (0.00150 ± 0.00025), PNG (0.00108 ± 0.00024), Thailand (0.00084 ± 0.00006), Vietnam (0.00061 ± 0.00010), and most conserved in Malaysia (0.00045 ± 0.00011). Overall, nucleotide diversity was calculated 0.00097 ± 0.00005 ($\pi \pm$ S.D.) (Table 1). Neutrality selection test within intraspecies by Tajima's D (-2.10845), Fu and Li's D* (-5.95822), and Fu and Li's F* (-5.05392) showed a significant negative value and the ratio of $d_N/d_S$ (4.52) greater than one. All of calculation parameters strongly indicated a positive selection and population expansion of PvEBP-ecto (Table 1 and Table 2). The inter-species level was calculated by the robust MK test with *P. cynomolgi* DBP2 gene as an out-group species. The PvEBP-ecto analysis showed a neutrality index as 2.274 ($p = 0.051$) indicating positive selection however this did not reach statistical significance (Table 2). The PvEBP-RII domain showed a significantly strong positive selection pressure and PvEBP-RIII-V domain showed negative/purifying selection that did not reach statistical significance (Table 2).

*Pvebp* sequences identified 67 haplotypes with four distinct clusters. The cluster 1 contained 61 haplotypes with geographically sharing for eight countries (Fig 2). The result indicates that the *pvebp* cluster 1 polymorphism occur widely without specific geographical pattern. The far distance from core haplotypes population were revealed specific geographical pattern from

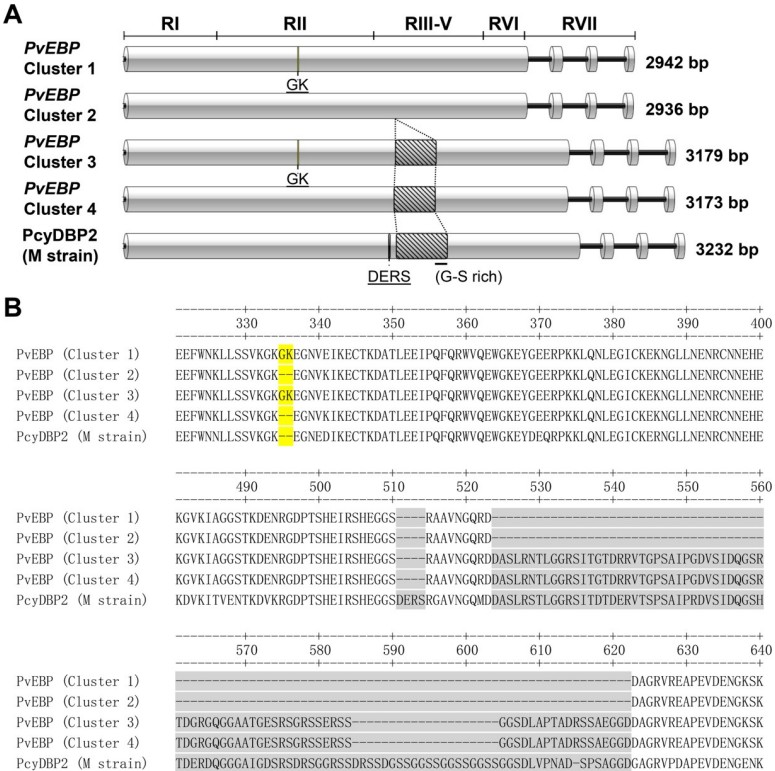

**Fig 1. *Pvebp* primary structure and phenotypic clustering.** (A) Disc shape shows *pvebp* exon and solid line represents intron part. Region separation was followed by PvDBP region divided strategy on cluster 1. The line in the box represents indels variation sites (GK or DERS). Cluster 3, 4, and PcyDBP2 (M strain) sequences have large insertions (oblique line box) in RIII-V domain. (B) Amino acid sequence alignment with PvEBP cluster shows in yellow background for Gly-Lys (GK), and gray background shows large indels variation.

Ethiopia for cluster 2 (H51), cluster 3 (H46, H50, and H51), and ROK isolates for cluster 4 (H01 and H02) (Fig 2).

*Pvebp*-domain base nucleotide diversity analysis showed in RII (0.00172 ± 0.00009) followed by RI (0.00080 ± 0.00010), RVI (0.00062 ± 0.00010), and most conserved in RIII-V (0.00018 ± 0.00009) (Fig 3, S1 Table).

**Table 1. Estimates of nucleotide diversity, haplotype diversity and neutrality indices of *pvebp*-ecto domain based on the geographical location.** (\*$p<0.05$, \*\*$p<0.02$, \*\*\*$p<0.01$).

| Location | No. of samples | SNPs | No. of haplotype | Diversity ± S.D. Haplotype (Hd) | Diversity ± S.D. Nucleotide ($\pi$) X $10^3$ | Tajima's D | Fu and Li's D* | Fu and Li's F* |
|---|---|---|---|---|---|---|---|---|
| China | 4 | 8 | 4 | 1.000 ± 0.177 | 1.79 ± 0.46 | -0.44637 | -0.44637 | -0.43935 |
| Ethiopia | 24 | 16 | 14 | 0.938 ± 0.030 | 1.72 ± 0.11 | -0.24556 | 0.16286 | 0.04577 |
| Malaysia | 53 | 8 | 6 | 0.368 ± 0.083 | 0.45 ± 0.11 | -1.08130 | -0.14673 | -0.52964 |
| Myanmar | 10 | 9 | 6 | 0.889 ± 0.075 | 1.50 ± 0.25 | 0.42326 | 1.02910 | 0.98872 |
| PNG | 16 | 12 | 7 | 0.833 ± 0.072 | 1.08 ± 0.24 | -1.17924 | -0.81090 | -1.05049 |
| ROK | 10 | 9 | 5 | 0.844 ± 0.080 | 1.69 ± 0.16‘ | 1.00440 | 0.62312 | 0.80625 |
| Thailand | 174 | 33 | 37 | 0.845 ± 0.024 | 0.84 ± 0.06 | -1.92027* | -3.62687** | -3.50982** |
| Vietnam | 25 | 7 | 8 | 0.773 ± 0.070 | 0.61 ± 0.10 | -0.74586 | -0.71112 | -0.83830 |
| **Overall** | **316** | **53** | **67** | **0.823 ± 0.021** | **0.97 ± 0.05** | **-2.10845*** | **-5.95822**** | **-5.05392**** |

**Table 2. McDonald-Kreitman (MK) tests on PvEBP with PcyDBP2 as out-group species and $d_N/d_S$ ratio.**

| PvEBP domain | Polymorphic changes within *P. vivax* | | Fixed differences between *P. vivax* and *P. cynomolgi* | | Neutrality index (*p* value)[a] | $d_N/d_S$ |
|---|---|---|---|---|---|---|
| | Syn | Non-Syn | Syn | Non-Syn | | |
| PvEBP-ecto | 8 | 45 | 57 | 141 | 2.274 (0.051) | 4.52 |
| PvEBP-RI | 3 | 8 | 16 | 40 | 1.067 (1.000) | 0.67 |
| PvEBP-RII | 1 | 25 | 23 | 37 | 15.541 (0.001) | 27.61 |
| PvEBP-RIII-V | 17 | 33 | 15 | 43 | 0.677 (0.402) | 0.75 |
| PvEBP-VI | 0 | 2 | 5 | 17 | - | - |

[a] Fisher's exact test *p*-value.

## Antigenicity screening

PvEBP-RII (30.1 kDa) and PvEBP-RIII-V (25.8 kDa) recombinant proteins were used for antigenicity evaluation (Fig 4A). Total IgG reactivity in vivax patient showed significantly higher in RII (MFI ± S.D., 16,048 ± 8,547) than RIII-V (7,386 ± 3,211). Additionally, RII IgG reactivity in healthy individual also showed significantly higher in RII (11,175 ± 5,010) than RIII-V (5,462 ± 1,762) (Table 3). However, PvEBP specific IgG inducing level after *P. vivax* infection

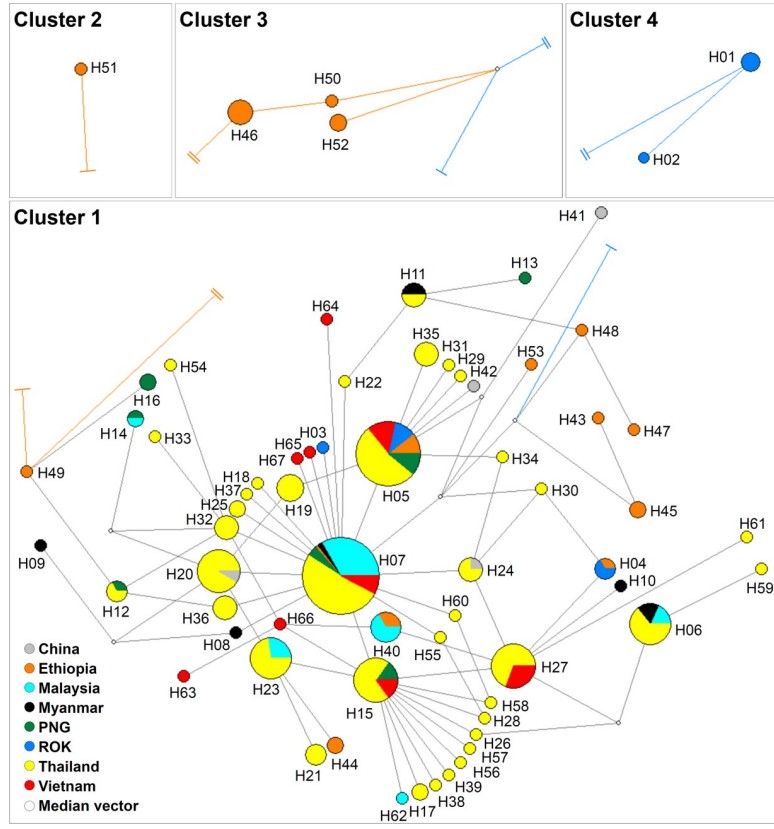

**Fig 2. Median-joining networks of PvEBP-ecto haplotype.** The geographical haplotype network shows the relationships among 67 haplotypes present in 316 isolates sequence. Distances between nodes are generated by NetWork 5.0 software. The orange cutting line connected to cluster 2 and cluster 3, the blue cutting line connected to cluster 4. The cluster 3 and cluster 4 contained large insertion sequence in RIII-V domain and cluster 4 connected to PcyM DBP2.

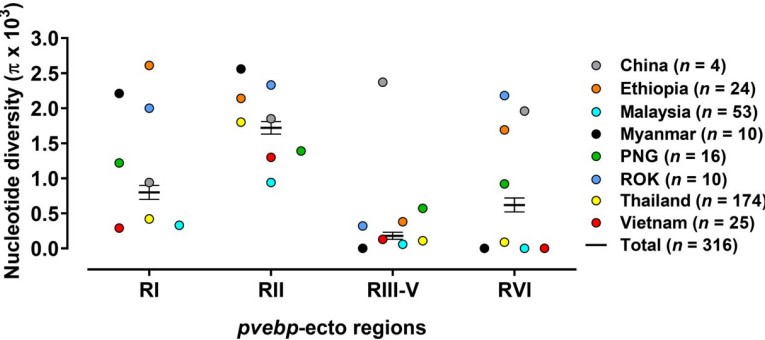

**Fig 3. *Pvebp* domains nucleotide diversity (π) based on the geographical areas.** PvEBP region divided for region I (RI, 1–534 bp), RII (535–1,437 bp), RIII-V (1,438–2,070 bp), and RVI (2,071–2,307 bp) based on the PvDBP homologue region.

in RIII-V (21.5%) was higher than RII (16.1%) (Table 3). This result indicated that RII antigen mainly recognized by natural IgG for innate protection, whereas specific IgG inducing level by B-cell was lower than RIII-V domain. On the other hand, RIII-V had lower level of natural IgG reactivity caused by neutral selection for immune evasion, whereas PvEBP specific IgG production level was higher than RII after infection (Fig 4B and Table 3). Antigenicity correlation analysis between RII and RIII-V in individual vivax patient showed significant correlation from ROK ($r = 0.710$), Thailand ($r = 0.534$), and Myanmar ($r = 0.305$) (Fig 4C). Combination of all countries correlation between RII and RIII-V showed low level of significant positive correlation ($r = 0.417$) (Fig 4). The correlation analysis of RII and RIII-V with age and parasitaemia showed positively correlation only in RIII-V with patient age (Fig 5A–5D).

## Discussion

This study describes the genetic diversity of the extracellular domain (ecto) of *P. vivax* erythrocyte binding protein (*pvebp*) genes from eight countries and correlated these polymorphisms with vivax patient antigenicity from different endemic areas. The genetic phenotypes of *pvebp*-ecto were classified into four clusters, based on the insertion/deletion (indels) variation. Generally, the gene indels variation of pathogen directly affects host antibody recognition. Cluster 1 is a major genetic phenotype of *pvebp* which widely shares its geographical haplotype. However, three minor clusters (cluster 2, 3, and 4) which provide geographically distinct patterns.

Interestingly, the large insertion sequence in PvEBP-RIII-V domain of cluster 3 and 4 was conserved in all of the tested field isolates regardless of geographical location. This indels variation is interesting to note that PvEBP cluster 4 phenotype is closely related with *P. cynomolgi* Mulligan strain DBP2 (PcyM DBP2) which has a Gly-Lys (GK) amino acid deletion in EBP-RII domain following the large insert in EBP-RIII-V domain. Thus, PcyM DBP2 gene is clearly an orthologue of PvEBP. Recently, *P. cynomolgi*, has been used as a model to study the invasion biology of *P. vivax* [34, 35]. The EBL family in *P. vivax* has two genes named PvDBP and PvEBP, whereas *P. cynomolgi* has three genes named PcyDBP1, PcyDBP2, and PcyEBP (PcyM DBP2) [36]. The additional Gly- and Ser-rich sequence insertion in both *P. vivax* and *P. cynomolgi* EBP genes may have a function as a linker that possibly affect flexibility for inducing an action radius of ecto domain and hamper host antibodies recognition [37, 38]. However, the function of this insertion on PvEBP and PcyEBP (PcyM DBP2) needs to be determined.

The importance of EBL family proteins RIII-V domain as a vaccine candidate was suggested in the previous study of antibodies against *P. falciparum* erythrocyte binding antigen 175 (PfEBA-175) RIII-V domain which was associated with protection against symptomatic case

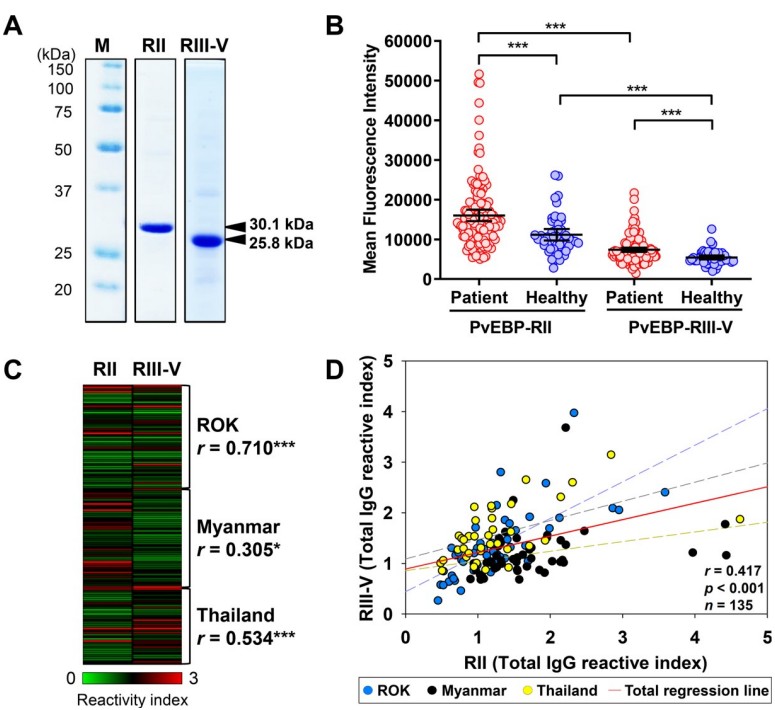

**Fig 4. Humoral immune response of PvEBP-RII and PvEBP-RIII-V.** (A) Purity confirmation by SDS-PAGE of recombinant PvEBP-RII (30.1 kDa) and PvEBP-RIII-V (25.8 kDa) expression. (B) Total IgG prevalence of each domain with the vivax patient (red dot) and healthy individual (blue dot) sera. The bar indicates the mean fluorescence intensity (MFI) ± 95% CI. The *p* values were calculated by Student's *t*-test. Significant differences are shown as triple asterisks *p* <0.001. (C) IgG prevalence visualized for comparison between RII and RIII-V with each patient sera by normalized reactivity index. Significant differences are shown as single asterisks *p* <0.05 and triple asterisks *p* <0.001. (D) Correlation between RII and RIII-V total IgG reactive indices using Pearson correlation test (*r*). Blue dot and dash line represent patient sera reactive index from ROK and its regression line. Black and yellow dot and dash line represent reactivity indices and its regression lines from Myanmar and Thailand patient sera, respectively. Red line indicates total regression.

**Table 3. Humoral immune responses of PvEBP-RII and -RIII-V domains.**

| Antigen/location | No. of patient samples | | | 95% CI[b] | MFI[c] | No. of healthy samples | | | 95% CI | MFI | *p* value[e] |
|---|---|---|---|---|---|---|---|---|---|---|---|
| | Positive | Negative | Total (%)[a] | | | Positive | Negative | Total (%)[d] | | | |
| **RII** | **22** | **115** | **137 (16.1)** | 10.8–23.1 | 16048.0 | **2** | **48** | **50 (96.0)** | 86.5–98.9 | 11175.3 | 0.0002 |
| ROK | 5 | 45 | 50 (10.0) | 4.4–21.4 | 14092.0 | | | | | | |
| Myanmar | 12 | 38 | 50 (24.0) | 14.3–37.4 | 19315.7 | | | | | | |
| Thailand | 5 | 32 | 37 (13.5) | 5.9–28.0 | 14275.5 | | | | | | |
| **RIII-V** | **29** | **106** | **135 (21.5)** | 15.4–29.2 | 7386.3 | **2** | **48** | **50 (96.0)** | 86.5–98.9 | 5462.2 | < 0.0001 |
| ROK | 14 | 36 | 50 (28.0) | 17.5–41.7 | 7384.6 | | | | | | |
| Myanmar | 3 | 45 | 48 (6.3) | 2.6–16.8 | 6460.0 | | | | | | |
| Thailand | 12 | 25 | 37 (32.4) | 19.6–48.5 | 8590.5 | | | | | | |

[a]Sensitivity: percentage of positive in patient samples.

[b]CI: confidence interval.

[c]MFI: mean fluorescence intensity.

[d]Specificity: percentage of negative in healthy samples.

[e]Differences in the total IgG prevalence for each antigen between vivax patients and healthy individuals were calculated with Mann-Whitney *U*-test. A *p* value of < 0.05 is considered statistically significant.

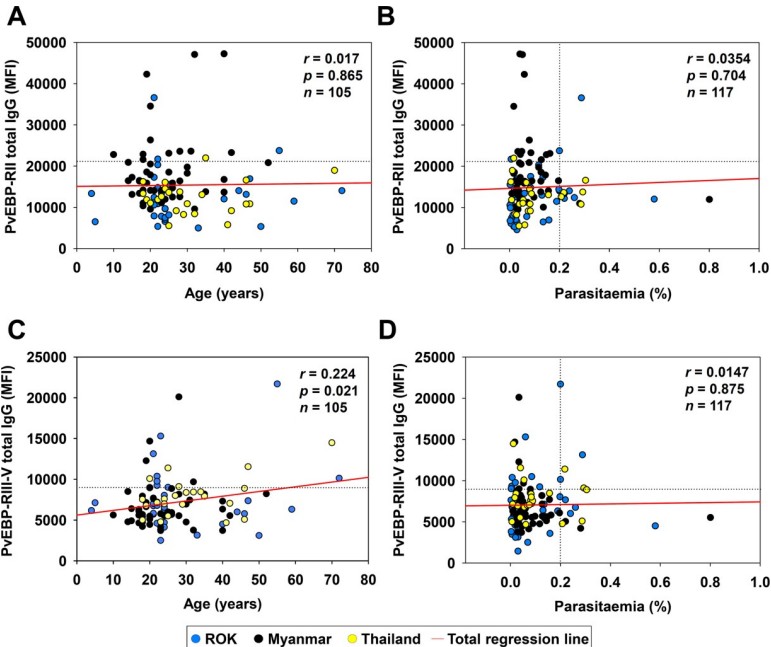

**Fig 5. The correlation of parasitaemia and age with PvEBP domains.** (A and B) PvEBP-RII and (C and D) PvEBP-RIII-V total IgG indices obtained from mean fluorescence intensity (MFI) were evaluated correlation with patient age (years) and parasitaemia (%) using Pearson correlation test (*r*), respectively. The horizontal dash line indicates MFI+2S.D. value as positive reactivity and vertical dash line in parasitaemia (>0.2) considered high parasitaemia.

of malaria [39]. PfEBAs (PfEBA-175, PfEBA-140, and PfEBA-181) RIII-V domain was dominantly eliciting IgG1 and IgG3 by patient age-dependent manner [39]. The function of IgG subtype in immune strategy of malaria infection is well documented, in the opsonisation capacity of IgG1 and IgG3 [8]. Similarly, an earlier study demonstrated that an IgG1 response against PvEBP domain (161–641 aa.) was dominant in an age-dependent manner in vivax patients [40]. Our study confirmed that only the IgG response against PvEBP-RIII-V (480–676 aa.) domain was positively correlated with patient age. Thus even though PvEBP-RIII-V domain has large sequence insertion and shows low frequency in the tested isolates, the RIII-V domain needs to determine its function and considered for vaccine design in the future.

The EBL (RII) domain in *Plasmodium* species is the key domain for host cell binding and invasion by interacting with host cell glycoprotein receptor [11]. Therefore, this functional domain of PvEBP-RII is the most logical target for a parasite-erythrocyte invasion blocking vaccine [41]. Antibodies against PvEBP-RII were shown to prevent the binding of a recombinant protein encoding this domain to reticulocytes [21]. Additionally, antibodies against PvEBP-RII were useful for serological marker of recent malaria exposure, with a half-life estimated at 734 days [42]. Anti-PvEBP-RII antibodies were predominant IgG1 and IgG3 which is similar to other PfEBAs-RII to mediate opsonic phagocytosis [39, 40]. However, PvEBP-RII showed highest genetic polymorphism within *pvebp*-ecto domains. This high polymorphic pattern is consistent with other EBL members in *P. vivax* and *P. falciparum* [11, 41] and its function as binding domains has been conserved in the other members of the EBL family [43]. As high polymorphisms, PvDBP-RII domain presented difficulty in developing an efficacious vaccine [44, 45]. It would be important to consider the genetic polymorphisms of PvEBP-RII when developing a vaccines targeting this domain [17].

Study of PvEBP copy number variation (CNV) revealed higher multiple copy ratios in Madagascar (56%) than in Cambodian (19%) isolates, possibly correlating with the Duffy dependence phenotype [46]. Geographically, PvEBP gene CNV pattern was similar to PvDBP which detected higher CNV in Africa region such as Ethiopia (79%) and Madagascar (46%) than South-East Asia such as Thailand (30%), Cambodia (28%), Indonesia (6%), and Malaysia (4%) [26, 46]. In this study, Ethiopia isolates covered three different clusters (clusters 1, 2, and 3) which indicate high level of genetic and phenotypic diversity in the African region where the population is largely Duffy-negative [47]. The Duffy negativity may have affected both *pvebp* and *pvdbp* gene-phenotype to change according to host cell preference or/and immune evasion mechanism [23]. The neutrality test of *pvebp*-ecto showed that rare alleles were present at high frequencies. Especially, PvEBP-RIII-V which showed evidence of population expansion and a negative/purifying selection effect. This was in contrast to the PvEBP-RII domain which showed positive selection pressure. Previous reports of PvEBP-RII neutral selection revealed positive/diversifying selection similar to other EBL domains in human and non-human primate malaria [46, 48, 49]. All of these results support that adaptation to the host environment lead both PvEBP-RII and PvEBP-RIII-V domains to be highly variable.

The antigenicity screening result also reflects the influence of a high level of a PvEBP gene variant. Although both PvEBP-RII (16.1%) and PvEBP-RIII-V (21.5%) domains significantly induced antibody responses in vivax infected patients, these responses were comparatively lower than other *P. vivax* antigens [50–52]. Previous studies on vivax malaria patients, (using the same antigenicity screening method used in this study) conveyed the antibody responses to be PvDBP-RII (56.9%), PvRBP1a-34 (33.7%), PvRBP1b-32 (39.4%), and PvGAMA-ecto (72.0%) which were localized at apical organelle and important for host cell interaction [51, 52]. Relatively low antigenicity of PvEBP could be a result of high polymorphism with positive selection and population expansion in the PvEBP-RII domain, and purifying selection in PvEBP-RIII-V domain. Interestingly, total IgG production and recognition of PvEBP-RII and PvEBP-RIII-V showed clearly distinct properties. The total IgG response against PvEBP-RII was higher than PvEBP-RIII-V, however, acquired antibody response was less than PvEBP-RIII-V. In contrast, PvEBP-RIII-V elicited higher IgG responses in vivax patients, yet, basal IgG recognition level was poor.

PvEBP is a prominent novel vaccine candidate for blood-stage *P. vivax* malaria with its potential to target both Duffy-dependent and independent *P. vivax* parasites. However, low antigenicity due to high genetic polymorphism in the PvEBP-RII, and antigen phenotype and selection pressure in the PvEBP-RIII-V will need to be considered for future vaccine development.

## Supporting information

**S1 Fig. PvEBP-ecto gene amplification from ROK isolates.** PvEBP-ecto gene in ROK samples shown different amplicon size for approximately 2,556 bp for K_01, 02, 04, and 05 and 2,325 bp for K_03, 06, 07, 08, 09, and 10.
(TIF)

**S1 Table. Estimates of nucleotide diversity, haplotype diversity and neutrality indices of PvEBP each domain based on the location.** ($p<0.05$, $^{**}p<0.02$, $^{***}p<0.01$)
(XLSX)

## Acknowledgments

The authors are grateful for all the staff and patients associated with the Shoklo Malaria Research Unit for *P. vivax* sample donation (SMRU, Thailand) and clinics and patients from ROK, Myanmar, China and PNG.

## Author Contributions

**Conceptualization:** Jin-Hee Han.

**Data curation:** Sarah Auburn.

**Formal analysis:** Jin-Hee Han, Jee-Sun Cho, Jessica J. Y. Ong.

**Funding acquisition:** Bruce Russell, Eun-Taek Han.

**Investigation:** Jee-Sun Cho.

**Methodology:** Jin-Hee Han.

**Project administration:** Jin-Hee Han, Jee-Sun Cho, Laurent Rénia, Bruce Russell, Eun-Taek Han.

**Resources:** Jin-Hee Han, Jee-Sun Cho, Ji-Hoon Park, Myat Htut Nyunt, Edwin Sutanto, Hidayat Trimarsanto, Beyene Petros, Abraham Aseffa, Sisay Getachew, Kanlaya Sriprawat, Nicholas M. Anstey, Matthew J. Grigg, Bridget E. Barber, Timothy William, Gao Qi, Yao-bao Liu, Richard D. Pearson, Sarah Auburn, Ric N. Price, Francois Nosten, Eun-Taek Han.

**Software:** Jin-Hee Han, Richard D. Pearson.

**Supervision:** Laurent Rénia, Bruce Russell, Eun-Taek Han.

**Validation:** Jin-Hee Han, Jee-Sun Cho, Laurent Rénia.

**Visualization:** Jin-Hee Han, Jessica J. Y. Ong.

**Writing – original draft:** Jin-Hee Han.

**Writing – review & editing:** Jin-Hee Han, Jee-Sun Cho, Jessica J. Y. Ong, Sarah Auburn, Ric N. Price, Laurent Rénia, Bruce Russell, Eun-Taek Han.

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
