## [Decision Letter · Decision Letter 0]

12 Dec 2019

Dear Dr. HAN:

Thank you very much for submitting your manuscript "Genetic diversity and neutral selection in Plasmodium vivax erythrocyte binding protein correlates with patient antigenicity" (PNTD-D-19-01925) for review by PLOS Neglected Tropical Diseases. Your manuscript was fully evaluated at the editorial level and by independent peer reviewers. The reviewers appreciated the attention to an important topic but identified some aspects of the manuscript that should be improved.

We therefore ask you to modify the manuscript according to the review recommendations before we can consider your manuscript for acceptance. Your revisions should address the specific points made by each reviewer.

(1) A letter containing a detailed list of your responses to the review comments and a description of the changes you have made in the manuscript.

(2) Two versions of the manuscript: one with either highlights or tracked changes denoting where the text has been changed (uploaded as a "Revised Article with Changes Highlighted" file ); the other a clean version (uploaded as the article file).

(3) If available, a striking still image (a new image if one is available or an existing one from within your manuscript). If your manuscript is accepted for publication, this image may be featured on our website. Images should ideally be high resolution, eye-catching, single panel images; where one is available, please use 'add file' at the time of resubmission and select 'striking image' as the file type. 

Please provide a short caption, including credits, uploaded as a separate "Other" file. If your image is from someone other than yourself, please ensure that the artist has read and agreed to the terms and conditions of the Creative Commons Attribution License at http://journals.plos.org/plosntds/s/content-license (NOTE: we cannot publish copyrighted images). 

(4) Appropriate Figure Files 

Please remove all name and figure # text from your figure files upon submitting your revision. Please also take this time to check that your figures are of high resolution, which will improve both the editorial review process and help expedite your manuscript's publication should it be accepted. Please note that figures must have been originally created at 300dpi or higher. Do not manually increase the resolution of your files. For instructions on how to properly obtain high quality images, please review our Figure Guidelines, with examples at: http://journals.plos.org/plosntds/s/figures

While revising your submission, please upload your figure files to the Preflight Analysis and Conversion Engine (PACE) digital diagnostic tool, https://pacev2.apexcovantage.com/ PACE helps ensure that figures meet PLOS requirements. To use PACE, you must first register as a user. Then, login and navigate to the UPLOAD tab, where you will find detailed instructions on how to use the tool. If you encounter any issues or have any questions when using PACE, please email us at figures@plos.org.

We hope to receive your revised manuscript by Feb 10 2020 11:59PM. If you anticipate any delay in its return, we ask that you let us know the expected resubmission date by replying to this email.

To submit your revised files, please log in to https://www.editorialmanager.com/pntd/

Sincerely,

Joseph M. Vinetz

Deputy Editor

Joseph Vinetz

Deputy Editor

Reviewer's Responses to Questions

**Key Review Criteria Required for Acceptance?**

**Methods**

-Are the objectives of the study clearly articulated with a clear testable hypothesis stated?

-Is the study design appropriate to address the stated objectives?

-Is the population clearly described and appropriate for the hypothesis being tested?

-Is the sample size sufficient to ensure adequate power to address the hypothesis being tested?

-Were correct statistical analysis used to support conclusions?

-Are there concerns about ethical or regulatory requirements being met?

Reviewer #1: I suggest to comment more about the neutrality tests, sometimes significant negative values simply indicate population expansion rather than purifying selection (see Molecular biology and evolution 30: 2050-2064 and similar types of work). I also suggest some form of MK tests to be applied, together with estimating Ds Dn. I believe that a more detailed comparison of different neutrality tests will benefit this work.

**Results**

-Does the analysis presented match the analysis plan?

-Are the results clearly and completely presented?

-Are the figures (Tables, Images) of sufficient quality for clarity?

Reviewer #1: This is an interesting report. Usually immunity is related to balancing selection and the authors did not find evidence of it using these neutrality tests. Please provide the sequences accession numbers.

**Conclusions**

-Are the conclusions supported by the data presented?

-Are the limitations of analysis clearly described?

-Do the authors discuss how these data can be helpful to advance our understanding of the topic under study?

-Is public health relevance addressed?

Reviewer #1: Need to discuss more the limitations of neutrality tests in this context.

**Editorial and Data Presentation Modifications?**

Reviewer #1: Minor revision (need to deposit data in the GenBank or other major database).

**Summary and General Comments**

Reviewer #1: This is an interesting report. Usually immunity is related to balancing selection and the authors did not find evidence of it using these neutrality tests. However, I suggest to comment more about it, sometimes significant negative values simply indicate population expansion rather than purifying selection (see Molecular biology and evolution 30: 2050-2064 and similar types of work). I also suggest some form of MK tests to be applied, together with estimating Ds Dn. I believe that a more detailed comparison of different neutrality tests will benefit this work. The accession numbers of the sequences should be provided.

PLOS authors have the option to publish the peer review history of their article (what does this mean?). If published, this will include your full peer review and any attached files.

Reviewer #1: No

---

## [Editor Report · Decision Letter 1]

8 Mar 2020

Dear Dr. HAN,

We are pleased to inform you that your manuscript 'Genetic diversity and neutral selection in Plasmodium vivax erythrocyte binding protein correlates with patient antigenicity' has been provisionally accepted for publication in PLOS Neglected Tropical Diseases.

Best regards,

Joseph M. Vinetz

Deputy Editor

Joseph Vinetz

Deputy Editor

---

## [Editor Report · Acceptance letter]

19 Jun 2020

Dear Dr. HAN,

We are delighted to inform you that your manuscript, "Genetic diversity and neutral selection in *Plasmodium vivax* erythrocyte binding protein correlates with patient antigenicity," has been formally accepted for publication in PLOS Neglected Tropical Diseases.

Best regards,

Shaden Kamhawi

co-Editor-in-Chief

Paul Brindley

co-Editor-in-Chief
